# A Fenchone Derivative Effectively Abrogates Joint Damage Following Post-Traumatic Osteoarthritis in Lewis Rats

**DOI:** 10.3390/cells11244084

**Published:** 2022-12-16

**Authors:** Idan Carmon, Reem Smoum, Eli Farhat, Eli Reich, Leonid Kandel, Zhannah Yekhtin, Ruth Gallily, Raphael Mechoulam, Mona Dvir-Ginzberg

**Affiliations:** 1Multidisciplinary Center for Cannabinoid Research, Hebrew University of Jerusalem, Jerusalem 9112102, Israel; 2Institute of BioMedical and Oral Research, Faculty of Dental Medicine, Hebrew University of Jerusalem, Jerusalem 9112102, Israel; 3The Institute for Drug Research, School of Pharmacy, Faculty of Medicine, Hebrew University of Jerusalem, Jerusalem 9112102, Israel; 4Orthopedic Surgery Complex, Hebrew University-Hadassah Medical Center, Jerusalem 91120, Israel; 5The Lautenberg Center for Immunology and Cancer Research, Faculty of Medicine, The Hebrew University of Jerusalem, Jerusalem 9112102, Israel

**Keywords:** fenchone derivatives, cartilage, osteoarthritis, anabolism, joint structure, DMOAD

## Abstract

Background: In a previous report, we have identified the cannabinoid receptor 2 (CB2) agonist HU308 to possess a beneficial effect in preventing age and trauma-induced osteoarthritis (OA) in mice. The effects of HU308 were largely related to the capacity of this compound to induce cartilage anabolism which was dependent on the CREB/SOX9 axis, and exhibited pro-survival and pro-proliferative hallmarks of articular cartilage following treatment. Here, we utilized the novel cannabinoid-fenchone CB2 agonists (1B, 1D), which were previously reported to render anti-inflammatory effects in a zymosan model. Methods: Initially, we assessed the selectivity of CB2 using a Gs-protein receptor cAMP potency assay, which was also validated for antagonistic effects dependent on the Gi-protein receptor cAMP pathway. Based on EC50 values, 1D was selected for a zymosan inflammatory pain model. Next, 1D was administered in two doses intra-articularly (IA), in a post-traumatic medial meniscal tear (MMT, Lewis rats) model, and compared to sham, vehicle, and a positive control consisting of fibroblast growth factor 18 (FGF18) administration. The histopathological assessment was carried out according to the Osteoarthritis Research Society International (OARSI) guidelines for rat models following 28 days post-MMT. Results: The G protein receptor assays confirmed that both 1B and 1D possess CB2 agonistic effects in cell lines and in chondrocytes. Co-administering a CB2 antagonists to 25 mg/kg 1D in a paw inflammatory pain model abolished 1D-related anti-swelling effect and partially abolishing its analgesic effects. Using an MMT model, the high dose (i.e., 24 µg) of 1D administered via IA route, exhibited reduced cartilage damage. Particularly, this dose of 1D exhibited a 30% improvement in cartilage degeneration (zonal/total tibial scores) and lesion depth ratios (44%), comparable to the FGF18 positive control. Synovitis scores remained unaffected and histopathologic evaluation of subchondral bone damage did not suggest that 1D treatment changed the load-bearing ability of the rats. Contrary to the anabolic effect of FGF18, synovial inflammation was observed and was accompanied by increased osteophyte size. Conclusion: The structural histopathological analysis supports a disease-modifying effect of IA-administered 1D compound without any deleterious effects on the joint structure.

## 1. Introduction

Our understanding of the role endocannabinoid signaling plays in cartilage biology remains quite limited. A previous report shows that the endochondral growth plate expresses both cannabinoid receptor 1 (CB1) and cannabinoid receptor 2 (CB2) in the hypertrophic chondrocyte layer and displays endocannabinoid tone, which affects limb growth during development [1]. Similarly, both receptors were found in adult cartilage [2], insinuating that the endocannabinoid tone is maintained throughout embryonic development till adulthood. In 2015, Sophocleous et al. showed a protective effect of HU308, a CB2 agonist [3], which was later reinforced by our research [4]. The lack of CB2 activity in advanced age was also a significant contributor to accelerated joint structural damage [3,4], indicating that CB2-related signaling is required for normal joint homeostasis and maintenance. Similarly, the upregulation of CB2 or its activation was also shown to exert anti-inflammatory effects on the synovium [5], which is in line with increased synovitis observed in CB2 null mice [4]. Significantly, these structural changes bestowed by CB2 agonists were also reported to show analgesic effects in various models, such as the monoiodoacetate-induced arthritis (MIA) model [6,7]. Cumulatively, the data also indicate that CB2 activation may render pro-anabolic effects to articular cartilage, potentially serving as disease-modifying OA drugs (DMOAD).

A recent development in cannabinoid-based compounds has brought forth a new generation of CB2 selective moieties, which have been established to possess biological anti-inflammatory actions [8]. These chemical entities were generated via coupling of (1S,4R)-(+) and (1R,4S)-(−)-fenchones with various resorcinols/phenols, which have been shown to possess biological effects that reduce pain behaviors, swelling and pro-inflammatory infiltrates in a zymosan model, in a dose-dependent manner [8]. Here, we attempted to characterize the selectivity, antagonistic, and agonistic activity of both 1D and 1B fenchone compounds based on G Protein-coupled Receptors (GPCR) pathway and subsequent cAMP intracellular response. Next, we examined the impact of 1D, which exhibited slightly better activity (lower EC50 levels) than 1B, was examined in a medial meniscal tear (MMT) rat model, which was subjected to histopathological and structural joint profiling.

## 2. Materials and Methods

### 2.1. Materials and Reagents

HU308, fenchones 1D and 1B were synthesized and characterized as previously reported [8,9] and formulated fresh before use in a solvent comprised of ethanol, cremophor (Kolliphor EL, Sigma-Aldrich, St Louis, MI, USA), and saline at 1:1:18 ratio, respectively. Initially, the required amount of 1D/1B was dissolved in ethanol, followed by the addition of cremophor with vigorous mechanical agitation to form a viscous micelle solution. Finally, the micelle solution was diluted in ice-cold saline for injection or culture assays. The vehicle solution consisted of ethanol, cremophor, and saline at a 1:1:18 volumetric ratio.

For intra-articular (IA) dosing, a 50 μL was injected into the tibiofemoral joint at two doses: 8 and 24 μg. As mentioned, vehicle control (1:1:18 ethanol:cremophor:saline) was utilized as placebo. Alternatively, Fibroblast growth factor 18 (FGF18; catalog No. 8988-F18-050, lot No. BVE0521071; R&D Systems, Minneapolis, MN) served as a positive control and was administered at 60 μg/mL via IA route (3 μg total). Notably, all animal procedures detailed below were carried out by Inotiv Boulder, according to the detailed procedures, which are also detailed below.

### 2.2. Agonist and Antagonist cAMP Secondary Messenger Assays

Cannabinoid receptors belong to G-protein-coupled receptors, which may transduce intracellular agonist (i.e., increased intracellular cAMP levels via Gs activation) or antagonist (i.e., decreased intracellular cAMP levels via Gi activation) responses. Fenchone derivatives were assayed using Eurofins proprietary assays (i.e., CB1-Antagonist Catalog # 86-0007P-2277AN; CB2-Antagonist Catalog ref. 86-0007P-2818AN; CB1-Agonist; Catalog # 86-0007P-2277AG; CB2-Agonist-Catalog # 86-0007P-2818AG; Luxembourg city, Luxembourg).

For agonist analysis, we monitored “evaluation of potency” (EC50) and % efficacy (Max response), which were reflected by increased intracellular cAMP levels detected in cAMP Hunter™ Gs cell lines (CHO-K1 lines) overexpressing human CNR1 (i.e., gene encoding CB1) or CNR2 (i.e., gene encoding CB2). Alternatively, antagonist activity (i.e., inhibitory concentration or IC50) utilized a cAMP Hunter™ Gi cell lines (CHO-K1 lines), detecting an intracellular reduction in cAMP levels in cells overexpressing CNR1 and CNR2. These cell lines are designed to be used in conjunction with the HitHunter^®^ cAMP Assay Detection Kit. Specifically, for agonist assay, cells were subjected to positive controls, known to activate the increase in cAMP for both receptors (agonist control 20 µM Forskolin for CB1 and 25 µM Forskolin for CB2). Alternatively, for antagonist assay, the cells were initially incubated with an agonist 20 μM or 25 μM CP55940, to agonize CB1 or CB2, respectively (Appendix A). After 30 min of incubation, the cells were incubated with a vehicle, a range of concentration for antagonist control (i.e., CB1 antagonist AM281 or CB2 antagonist SR144528), or a range of concentrations for 1D/1B fenchones. Prior to testing the cell, plating media were exchanged with 10 uL of Assay buffer (HBSS + 10 mM HEPES). Agonist results are expressed as percent efficacy relative to the maximum response of the control ligand (%Effc or EC50), while antagonist results are expressed as percent inhibition of the control ligand (IC50) normalized to the unantagonized vehicle.

### 2.3. Determining EC50 in Chondrocyte Cell Cultures

All human cell cultures were obtained from total knee replacement surgery (TKA) in accordance with Hadassah Medical Center Institutional Review Board approval and in accordance with the Helsinki Declaration ethical principles for medical research involving human subjects (Study # 0488-09). Following written informed consent, articular cartilage was obtained from the knee joints of OA patients undergoing total knee arthroplasty (*n* = 51, mean age 71 years, mean body mass index 31 kg/m^2^, Kellgren and Lawrence score ranging 3–4). Articular cartilage tissue was dissected, and chondrocytes were isolated and plated as described by Bar Oz et al. [10]. Isolated chondrocytes were passaged to passage 3 and plated in 96 well plates with DMEM media containing 10% FCS, 1% Penicillin-streptomycin, and 1% Amphotericin B. Cultures were maintained in standard incubation conditions (37 °C, 5% CO_2_) until confluence. All reagents for cell culture were purchased from Biological Industries (Beit-Haemek Kibutz, Israel) unless otherwise indicated. After confluence, chondrocytes were treated with 100 μM Forskolin (positive control, Sigma Aldrich, St Louis) and untreated cells for 45 min until inducing the reaction with the cAMP-Glo™ Assay, according to manufacturer’s instructions (Promega, Cat # V1501, Madison, WI, USA). Fenchones 1D, 1B, and HU308 were dissolved in DMSO and measured in 10^−5^–10^−12^ Molar concentration ranges. Luminescence was measured and a standard curve was simultaneously run per plate. ΔRLU was calculated by subtracting the RLU of the untreated sample from the RLU of the treated sample. Using this ΔRLU value and the linear equation generated from the standard curve, we calculate the cAMP concentration. Samples were normalized against maximum vs minimum average percentages and subjected to non-linear regression for stimulated dose-response via GraphPad to assess EC50 in treated human chondrocytes.

### 2.4. Animal Procedures

For the MMT model, male Lewis rats (*n* = 95 rats + 6 extra) were obtained from Envigo RMS, Inc. (Indianapolis, IN, USA), with a mean weight of 262 g. The animals were identified by a distinct mark at the base of the tail delineating group and animal number. After randomization, all cages were labeled with protocol number, group numbers, and animal numbers with appropriate color coding. During the acclimation and study periods, animals were housed in a laboratory environment with temperatures ranging from 19 °C to 25 °C and relative humidity of 30% to 70%. Automatic timers provided 12 h of light and 12 h of dark periods. Animals were allowed access ad libitum to Harlan Teklad Rodent Chow and fresh municipal tap water. Animal care, including room, cage, and equipment sanitation, conformed to the guidelines cited in the Guide for the Care and Use of Laboratory Animals (Guide, 2011) and the applicable Inotiv Boulder SOPs. Study protocols were approved by Inotiv IACUC standards.

For inflammatory pain assessment, nine to eleven-week-old female ICR mice were maintained in the Specific Pathogen Free (SPF) unit of the Hebrew University Hadassah Medical School, Jerusalem, Israel. The experimental protocols were approved by the Animal Care Ethical Committee of the Hebrew University-Hadassah Medical School, Jerusalem, Israel (#MD-20-16042-5). The animals were maintained on a standard pellet diet and water ad libitum. Mice were maintained at a constant temperature (20–21 °C) and a 12 h light/dark cycle.

### 2.5. Rat Model for Medial Meniscal Tear (MMT)

The rat OA model employed was surgically induced medial meniscal tear (MMT), wherein rats were anesthetized with Isoflurane (VetOne, catalog No. 502017, Boise, ID, USA), and the right knee area was prepared for surgery. A skin incision was made over the medial aspect of the knee, and the medial collateral ligament was exposed by blunt dissection and then transected. The medial meniscus was cut through the full thickness to simulate a complete tear. Skin and subcutis were closed with 4-0 Coated Vicryl (polyglactin 910) Violet Braided Suture (Ethicon, catalog No. J399H), and slight hand pressure was applied to the wound for approximately 3 min for hemostasis. Subcutaneous (SC) doses of buprenorphine (0.05 mg/kg) were administered after the animals awakened post-surgery. Rats were weighed daily from study day 1 through 7 and again on days 14, 21, and 28 (prior to necropsy). Vehicle was administered to two subgroups of the sham or MMT procedure, which served as controls (i.e., denoted as Sham, vehicle or MMT, vehicle, within figures and "sham" or "MMT" within the text). Dosing for vehicle and the fenchone 1D (24 μg-high and 8 μg-low concentrations) was initiated on study day 4, and thereafter on days 7, 10, 14, 17, 21, and 24. The fenchone dose was chosen after considering the previously injected IA concentration (4) to a mouse joint (0.5 μg) and extrapolating the dose 13-fold to a rats body weight and adding 20% surplus, reaching the lower 8 μg dose administered to the MMT rats. For FGF18 positive control (i.e., expected to exhibit anabolic cartilage response), dosing was initiated at day 7 post-MMT (60 μg/mL or 3 μg per knee). The animals were euthanized for necropsy 28 days post-surgery, following Isoflurane anesthesia, and bled to exsanguination followed by bilateral pneumothorax.

### 2.6. Post-Necropathy Histopathology Assessments

Right knees were collected from all animals and trimmed of muscle and patellae. The trimmed joints were placed in 10% neutral buffered formalin (NBF) for histologic processing and evaluation. Samples were decalcified for 21 days in 10% EDTA, pH = 7.5. Then, following dehydration in a graded series of ethanol washes, joints were embedded in paraffin and sectioned to 7 μm slices, following trimming off 1 mm until the tibiofemoral compartments were fully observed. Sections were obtained from each knee and stained with toluidine blue (0.04% in 0.2 M acetate buffer, pH = 4.0) based on a modified version of the methods used in Schmitz et al. (2010) [11] and Gerwin et al. (2010) [12]. The structural histological characterizations are detailed below.

#### 2.6.1. Medial Tibial Zonal Cartilage Degeneration Score

Regional differences across the tibial plateau were taken into consideration by dividing each section into three zones: (1) outside, (2) middle, and (3) inside. In the surgical OA model, the outside (zone-1) and middle (zone-2) thirds sre often the most severely affected, while milder changes are presented in the inside third (zone 3). Zones were scored individually and considered the area by which damage is evident (i.e. chondrocyte death/loss, proteoglycan (PG) loss, and collagen loss or fibrillation). Scoring ranged between 0–5, wherein “5” represented severe damage (criteria in SD3). A sum of all three zones was calculated and termed “Total tibial cartilage degeneration score”.

#### 2.6.2. Zonal Depth Ratio of Cartilage

The depth of any type of lesion (both chondrocyte and proteoglycan loss but may have good retention of collagenous matrix and no fibrillation) and the depth to tidemark are measured by an ocular micrometer at the midpoint in each of the 3 zones of the tibial plateau. A depth ratio of any matrix change is calculated by dividing the lesion depth by the total depth from the tidemark.

#### 2.6.3. Osteophyte Score and Measurement

Osteophyte thickness (tidemark to the furthest point extending toward synovium) was measured and scored according to a range from 0–5, as in criteria detailed in Appendix A.

#### 2.6.4. Medial Tibial Bone Damage and Sclerosis Scores

Damage to the calcified cartilage layer and subchondral bone was scored using the criteria in Appendix A. Generally, the damage was considered as fracturing or resorption of the calcified cartilage/subchondral bone with or without an invagination of deep zone cartilage into the subchondral bone layer.

#### 2.6.5. Synovitis Score

Synovial inflammation was scored (evaluation focuses on the lateral side since that is the area uncomplicated by the surgery) as indicated in Appendix A. Descriptions of other changes (typically fibrosis or acute inflammation/neutrophil infiltration extending into the lateral compartment usually associated with IA treatments) were also provided, if present.

### 2.7. Inflammatory Pain Assessment in Mice Paw

To induce inflammation, 40 μL of 1.5% (*w/v*) zymosan A (Sigma Aldrich) suspended in 0.9% saline was injected into the sub-planter surface of the right hind paw of the mice. Immediately after zymosan injection, CB2 antagonists dissolved in Phosphate-buffered saline were injected intraperitoneally (IP) and after 30 min, the fenchone derivatives (i.e., CB2 agonists) were dissolved in 0.1 mL vehicle containing 1:1:18 ethanol:cremophore:saline and injected IP. Control mice were injected with the vehicle only. After 2, 6, and 24 h, paw swelling, and pain perception were measured.

Specifically, measurement of edema formation and paw swelling was assessed by monitoring paw swelling via calibrated calipers (0.01 mm), at 6 and 24 h following injections of zymosan alone and/or the test compounds. Pain hyperalgesia was evaluated by the paw withdrawal von Frey test at 6 and 24 h following injections of zymosan and/or the test compounds. For the von Frey nociceptive filament assay, von Frey calibrated monofilament hairs of logarithmically incremental stiffness (0.008–300 g corresponding to 1.65–6.65 log of force) were used. In our study, only 1.4–60 g corresponding to 4.17 to 5.88 log of force was used to test the mouse sensitivity to a mechanical stimulus on a swollen paw. Notably, the measurements were performed in a quiet room. Before paw pain measurements, the animals were held for 10 s. The trained investigator applied the filament to the central area of the hind paw with a gradual increase in filament size. The test consisted of poking the middle of the hind paw to provoke a flexion reflex followed by a clear flinch response after paw withdrawal. Each one of the von Frey filaments was applied for approximately 3–4 s to induce the end-point reflex. The first test was carried out by using a force filament of 1.4 g. In case a withdrawal response was not detected, a higher stimulus was applied. The mechanical threshold force (in grams; g) was defined as the lowest force imposed by two von Frey monofilaments of various sizes required to produce a paw retraction. The untreated left hind paw served as a control.

### 2.8. Statistical Analysis

Group means and standard deviations (SD) were determined for each group. Treatment groups were compared to the vehicle disease control group (Vehicle-MMT) using a Kruskal-Wallis (KW) test with a Dunn’s post hoc analysis for scored (non-parametric) parameters. Sham control rats were compared to the disease control group using a Student’s Mann–Whitney U test (non-parametric). Statistical tests were performed using Prism version 9.3.0 software (GraphPad, 2010, San Diego, CA, USA). Statistical significance, according to Mann–Whitney or KW is denoted with an asterisk * *p* < 0.05, 2 asterisks ** *p* < 0.01, 3 asterisks *** *p* < 0.001, and 4 asterisks **** *p* < 0.0001. Graphical illustrations were carried out using BioRender software.

## 3. Results

### 3.1. Analysis of Fenchones 1D,1B Potency and Antagonistic Activity

As a first step, we assessed the capacity of 1D and 1B compounds to stimulate an agonistic effect by dose-dependent exposure of the compounds to cells either expressing the genes encoding CB1 (i.e., CNR1) or the gene encoding CB2 (i.e., CNR2), as compared to Forskolin positive control known to stimulate both receptors (Figure 1A). The results exhibit no response for CNR1 vs. a noticeable response for CNR2 (Appendix A, Figure 1A). Notably, EC50 for 1B was 0.05 μM vs. 0.01 μM for 1D, indicating superior potency of 1D. Next, we attempted to determine if there is an antagonistic effect of the 1D and 1B molecules, which may be a result of Gi activation and reduced intracellular cAMP. To this end, the antagonistic effects of AM281 and SR144528 were utilized as positive controls for CNR1 and CNR2-expressing cell lines, respectively (Appendix A, Figure 1B). The data show no detectable antagonistic effect of 1D or 1B for both receptors, indicating that the 1D/1B compounds are specifically agonistic to CNR2 and exert a Gαs effect, increasing intracellular cAMP levels. This was further confirmed in chondrocytes isolated from OA patients showing a low EC50 for 1B/1D vs. HU308, with a similar potency for both fenchone compounds (Figure 1C).

### 3.2. Assessment of Inflammatory Pain-Related Effects of 1D/1B Fenchones

We next attempted to utilize a mouse model to measure inflammatory pain, which is induced in mice paws via zymosan SC administration. Here, we used 1D since it exhibited lower EC50 levels and higher potency compared to 1B (Figure 1A). After administering 1D via IP route, mice were monitored for pain and swelling, as shown in the experimental setup presented in Figure 2A scheme. As a first step, we utilized a CB2 antagonist, AM630, to block CB2 activity and test if 1D exerts a CB2-dependent anti-inflammatory and/or anti-pain effect. The data in Figure 2B (Left graph) display reduced swelling at 6 h after administering 1D, which was maintained at 24 h. Adding AM630 to 1D, was able to reverse the anti-swelling effect of 1D, only 24 h after zymosan induction. These data were in line with the von Frey pain phenotypes showing a significant effect for 1D alone, which was partially reversed after blocking CB2 with AM630 at 24 h (Figure 2B, right graph). The pain data imply that 1D may reduce pain sensation, only in part through a CB2-dependent mechanism.

### 3.3. In Vivo Rat MMT Model

The rat MMT post-traumatic model was employed for the right limbs of Lewis rats, with all animals surviving study termination. Control groups consisted of sham or MMT procedures, each receiving vehicle intra-articularly (IA) at 4, 7, 10, 14, 17, 21, and 24 days, similar to the high and low doses of 1D. Notably, sham and MMT groups are indicated as “Sham, vehicle” or “MMT, vehicle” within figures. As positive DMOAD control, we utilized Fibroblast Growth Factor (FGF)-18 administered via IA route at 7, 14, and 21 days post-MMT. FGF18 was shown to bestow anabolic effect on articular cartilage [13] and has recently passed phase 2 for disease-modifying OA drug (DMOAD) [14].

Change in rat weight from 4 days to 28 days post-MMT appeared to be similar between the groups (Figure 3B). During the 28-day term, all treatments displayed higher gram force in the contralateral left joint vs. sham, which appeared to be most significantly increased in the FGF18-treated rats (Figure 3C). The cumulative area under the curves for dynamic weight bearing (DWB) exhibited the highest values in the FGF18-treated rats (Figure 3D) compared to all groups. While low dose 1D showed increased left to right load bearing vs. sham, high dose 1D appeared to have statistically insignificant load bearing vs. sham and vehicle-treated post-MMT (Figure 3D). That MMT vehicle control and shams did not exhibit statistical significance in the DWB area under the curve (Figure 3D) insinuate a possible placebo effect in this model. While the data do not suggest a dramatic analgesic effect of the high dose 1D, they highlight that its administration did not incur substantial pain effects to the joint, comparable to FGF18.

### 3.4. Histopathological Profiling of Post-MMT Joints

Following 28 days from MMT, joints were assessed for several structural hallmarks of OA. Figure 4A show the hallmarks assessed and their graphical illustration, which was largely based on Gerwin et al., 2010 [12]. In particular, the zonal and the sum of articular cartilage degeneration was assessed for the medial tibial joint compartment (a; Appendix A). Similarly, zonal depth ratios were determined based on the ratio of lesion depth vs. the expected depth of zone (b). For example, lesser damage would exhibit reduced zonal depth ratios. Synovitis scores were assessed according to the scoring table (c, Appendix A). Osteophyte measurements were manually taken, and osteophyte scores were assessed based on the table (d; Appendix A). Finally, tibial bone damage was scored based on the table (e; Appendix A). All raw scores are present in an excel sheet under Appendix A.

### 3.5. Total Tibial Cartilage Degeneration Score

Figure 4B,C show that MMT rats treated with vehicle had cartilage damage that was most severe in zone 1, and appeared to be reduced upon FGF18 administration. Similarly, FGF18 exhibited significantly reduced degeneration scores vs. all groups, with equivalent scores for 1D high dose. Interestingly, zones 1 and 2 appear to show a dose effect accompanied by reduced degeneration scores for the high dose of 1D. Finally, zone 3 appeared to show no beneficial effect of FGF18 treatment, yet 1D (i.e., for both high and low doses) exhibited lower degeneration scores for this zone, with statistical significance. The total zonal scores display significantly reduced degenerative scores for 1D high dose and FGF18 compared to the 1D low dose and vehicle groups post-MMT.

### 3.6. Medial Tibial Depth Ratio

Depth ratios exhibited similar trends as the degenerative scores, displaying significantly lower depth ratios for FGF18-treated rats, which were similar to the trend of the 1D high dose for zone 1 (Figure 5). Zone 2 exhibited a beneficial dose effect for the high dose of 1D, yet zone 3 displayed higher depth ratios for FGF18, which appeared to be significantly lower in high doses of 1D vs. vehicle, and FGF18. The mean zonal depth ratios exhibited reduced depth ratios for FGF18 and high dose 1D, while vehicle and low dose 1D exhibited similar mean depth ratios.

### 3.7. Assessments of Synovitis and Osteophyte Formation

As a next step, we determined synovitis scores ranging from 0 (i.e., normal synovial histopathology) to 5 (severe synovitis). The data show that medial synovial scores are highest for FGF18 vs. all groups (Figure 6A, arrows on representative images), possibly explaining the dynamic load-bearing profiles in these rats (Figure 3C,D). Interestingly synovitis scores were higher in the low dose 1D; however, the high dose 1D appeared similar to the sham and vehicle synovial scores. The data indicate that a high dose of 1D did not incur synovial inflammation, as observed upon FGF18 treatment.

Osteophyte measurements exhibited significantly higher measurements and scores for FGF18 vs. all groups (Figure 6B, arrows on representative images). While all groups exhibited higher scores and measurements for osteophytes compared to the sham group, the vehicle, 1D high and 1D low doses did not display any differences among them. The data do not support that 1D had prevented osteophyte formation compared to the vehicle; however, they did not exacerbate this structural change, as did FGF18.

### 3.8. Tibial Bone Damage

Bone damage is assessed according to the presence of basophilia at tidemark, causing loss of calcified cartilage or subchondral bone [12]. Another histological feature of such damage includes the presence of bone marrow lesions or articular cartilage collapse into the epiphysis beneath the tidemark (Appendix A, denoting a range of 0–5, wherein 5 is the most significant level of damage, Figure 7A illustration). Notably, all MMT groups exhibited increased collapse of deep zone cartilage into the subchondral bone site, depicted in the scores of MMT groups vs. sham (Figure 7B). Bone damage was slightly reduced in the 1D high dose and FGF18 (Figure 7B,C), albeit insignificant compared to all other post-MMT groups.

Cumulatively, the data support that high-dose 1D exhibited superior cartilage anabolism, which appeared to be similar to that of FGF18 positive control. Moreover, osteophyte and synovitis scores were lower fie 1D high dose vs. FGF18, potentially affecting the weight-bearing shifts observed in dynamic weight-bearing. Overall, the data appear to show a dose-dependent effect of IA administered 1D, which has equivalent DMOAD-like characteristics to FGF18, yet exhibited less deleterious structural effects to the synovium and bestowed preventative effects on inflammatory pain, as observed in the zymosan model.

## 4. Discussion

Our data highlight the potential beneficial effect of a new class of CB2 agonists on cartilage health and the potential prevention of OA. While intra-articular administration did not provide an analgesic effect noticeable in dynamic load bearing, it was less detrimental than FGF18. Moreover, both FGF18 and 1D exhibited striking improvement in the preservation of articular cartilage, as judged by the “Join degeneration scores” and “Depth ratio”. This structural effect was observed for 1D in a dose-dependent manner. Finally, the enhanced synovial inflammation, and osteophyte formation potentially affecting dynamic load bearing in FGF18-treated rats, was not observed with the high dose of 1D, indicating that it may not render any unwanted structural alteration that may affect load bearing and pain behaviors. Cumulatively, these data are in line with previous work by our group and others, which utilized HU308 to prevent OA [3,4,5]. CB2 ablation appeared to cause chondrocyte hypertrophy and may thus potentially contribute to osteophyte formation in OA. While we did not observe osteophytes in aging CNR2 nulls, using a post-traumatic model intra-articularly treated with HU308 reduced osteophyte formation associated with OA [3,4], which is in line with the data presented here with 1D.

The local use of the CB2 agonist IA for the treatment of OA was recently exposed and showed great promise [4]. It is justified as it does not cause adverse systemic effects and requires 200-fold lesser doses [4,15]. The potency effect of 1D/1B is superior over HU308 in chondrocytes, further accentuating that lower doses may be biologically efficacious. Particularly to the joint, the current formulation may be further efficacious due to the viscous nature of the intra-articular synovial fluid, which may cause retention of the compound in the joint to potentiate its biological action. In the mouse, for example, HU308 administration was found to reduce apoptosis and enhance SOX9 levels and PCNA, indicating a strong anabolic effect as a result of CB2 stimulation. Notably, both HU308 and 1D induce intracellular G-Protein activation in chondrocytes, which results in enhanced intracellular cAMP levels. We have shown that the rise in the cAMP levels may contribute to several CREB-responsive genes, one of which is SOX9. In previous work, CREB activation in osteoblasts by HU308 was shown to increase cyclin D1 and osteoblast proliferation [16], which is in line with our previous data [4]. Hence the local effect of such CB2 agonists may be powerful treatments in preventing OA structural decline and prolonged maintenance of joint function.

While pain-related benefits were not observed in our dynamic weight-bearing analysis, MMT rarely displays spontaneous alteration in weight-bearing, leaving the pain-related effects to be explored in other more severe pain models. In particular, MIA models appear to show a significant improvement in pain behaviors when administered with CB2 agonists, as the CB2 agonist JWH133 was reported to improve joint pain thresholds and dynamic weight bearing when applied systemically following MIA in mice [7]. Similarly, A-796260, a specific CB2 agonist, exhibited improved rat hind limb grip force and when applied systemically post MIA [17]. In fact, CB2 agonist HU308 has been shown to prevent synovial inflammation [5], which may activate synovial nociceptors in a neuro-immune axis [18,19,20]. Therefore, in models of mechanical joint trauma, the neuro-inflammatory axis may not be fully developed to result in profound baseline pain behaviors compared to other models as collagen-induced or MIA models [21]. Of note, HU308, as well as fenchones 1D and 1B, have shown significant improvement in preventing inflammatory pain in a zymosan model, which may be recapitulated in future pain models and pain behaviors. In summary, the use of CB2 agonists prevented joint damage, inflammation, and structural decline and may provide great promise as a novel DMOAD.

## 5. Conclusion and Limitations of Study

The data displayed in this report support the selective action of fenchone derivatives on CB2 and their enhanced potency in activating intracellular cAMP levels, as compared to HU308. In particular, 1D exhibited reduced swelling and pain phenotype, partially dependent on CB2, using a zymosan inflammatory pain model in mice. Moreover, the structural improvement after IA administration of 1D in a post-traumatic rat model supports its capacity to prevent joint damage to a comparable degree as FGF18. This structural effect of 1D was dose-dependent but did not improve dynamic weight bearing (DWB) after MMT vs. vehicle. On the other hand, the effects observed in DWB assays were not worsened upon 1D use vs. FGF18, which served as a DMOAD positive control. Summarizing the cumulative data support that 1D prevented post-traumatic structural joint damage, as well as inflammatory pain, in part by selectively targeting the activation of the CB2/cAMP axis.

Limitations of this study include the low number of mice subjected to the zymosan model of inflammatory pain. Another limitation involves pain phenotyping of the MMT model, which could have been extended to other behavioral analyses, such as static weigh bearing, Von Frey, and thermal plate response, to provide a more thorough phenotyping of pain.

## Figures and Tables

**Figure 1 cells-11-04084-f001:**
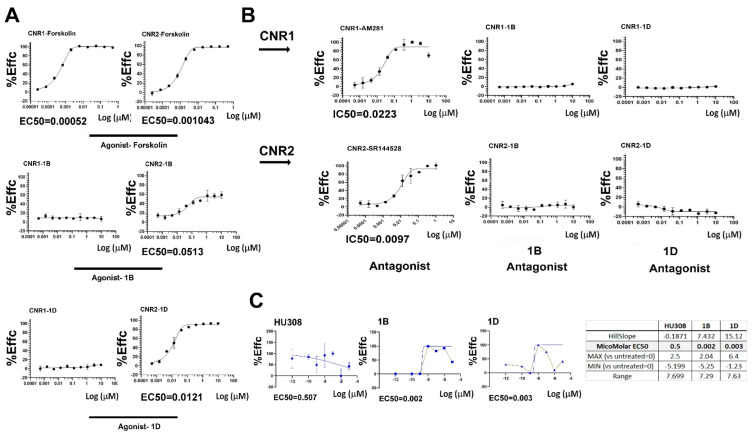
Fenchone 1D and 1B are selective agonists for CB2. (**A**) To test for agonist activity of GPCR cells expressing CNR1 or CNR2, cells were exposed to forskolin (positive agonist control), 1B and 1D fenchones, and assayed for EC50 values (μM), as in the Materials and Methods (Appendix A). X-axis display %Effc representing percent efficacy relative to the maximum response of the control ligand, while Y-axis display Log of μM concentration of ligand. (**B**) Antagonistic activity was assessed after CP55940 agonist induction, using a range of antagonist concentrations (AM281 for CNR1 and SR144528 for CNR2), as well as 1D and 1B fenchone compounds (Appendix A). IC50 values are determined by the cAMP inhibition levels observed after antagonist stimuli. (**C**) Human OA-derived articular chondrocytes were plated at confluence and treated with HU308, 1D and 1B fenchones for 45 min and assessed for EC50 values. The values were obtained using GraphPad Prism EC50 curve fits which display the minimum and maximum ranges along with the Hillslope fit value.

**Figure 2 cells-11-04084-f002:**
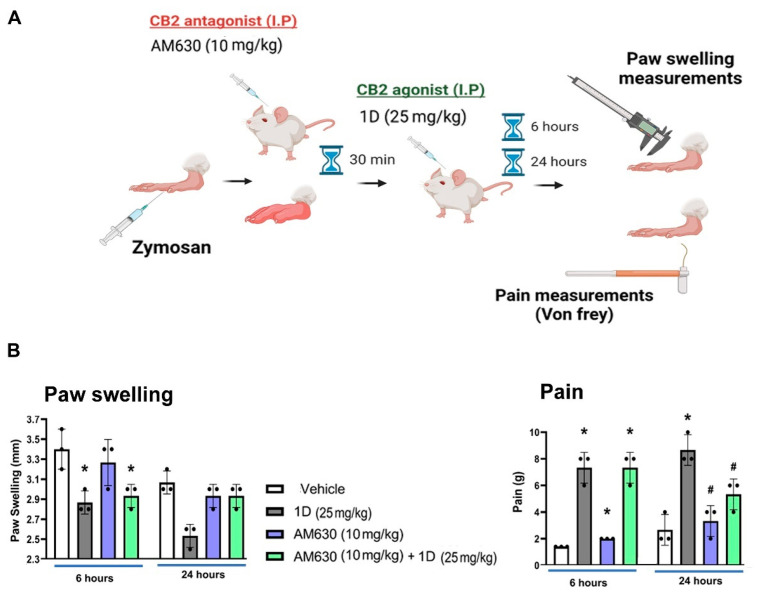
Assessment of 1D/1B effects in a mouse paw inflammatory pain model. Pain in mice was induced with zymosan and treated as detailed in materials and methods (**A**). (**B**) Graphs display paw swelling and pain (left and right graphs, respectively) of mice treated with 25 mg/kg (IP) 1D, and/or 10 mg/kg AM630. Statistical significance between treatments and vehicle control (denoted “*”) or between treatments and 1D control (denoted “#”), were examined via Kruskal–Wallis test with a Dunn’s post hoc analysis for scored (non-parametric) parameters, considering * *p* < 0.05 and # *p* < 0.05 to be statistically significant for both tests (*n* = 3).

**Figure 3 cells-11-04084-f003:**
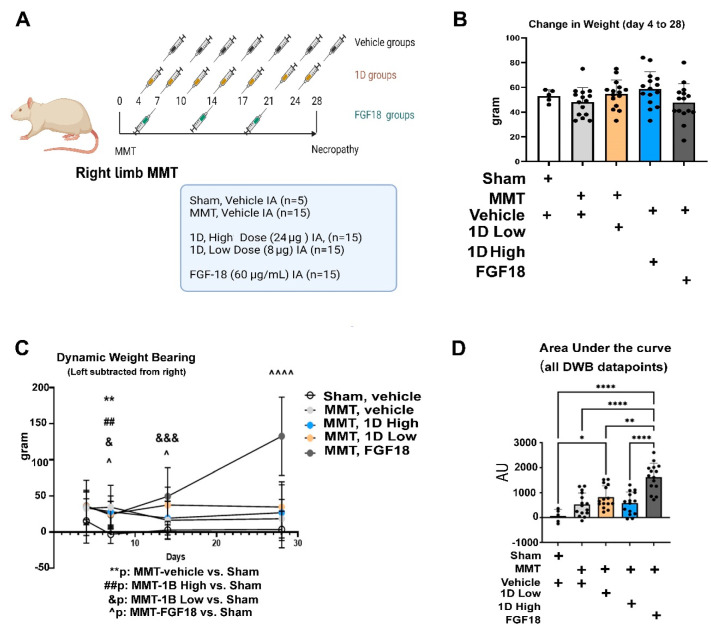
MMT experimental setup and behavioral phenotyping: (**A**) Experimental setup of MMT procedure carried out in the right hind limb of Lewis rats. Five groups were examined (Sham, *n* = 5; MMT-Vehicle *n* = 15; MMT-1D, low dose 8 μg *n* = 15; MMT-1D, high dose 24 μg *n* = 15; and MMT-FGF18 60 μg/mL or 3 μg *n* = 15). Vehicle and 1D groups were administered intra-articularly (IA) at 4, 7, 10, 14, 17, 21, and 24 days post-MMT, while the FGF18 group was administered IA at 7, 14, and 21 post-procedure. All rats were weighed (**B**) and subjected to dynamic weight bearing differences (left to right load bearing of hind limbs) (**C**) at 4, 7, 14, and 28 days post-MMT, prior to their sacrifice. (**D**) Area under the curve of dynamic weight bearing (DWB) between 4 and 28 days post-MMT. Statistical significance between treatments and control were examined via Kruskal–Wallis test with a Dunn’s post hoc analysis for scored (non-parametric) parameters. Comparisons of sham vs. Vehicle-MMT IA refer as *; Sham vs. 1B high refer as #; Sham vs 1B low refers as &; Sham vs. FGF18 refers as ^. considering one symbol (&, ^) *p* < 0.05 to be statistically significant. (**, ##,) *p* < 0.01; &&& *p* < 0.001; ^^^^ *p* < 0.0001. **** *p* < 0.0001.

**Figure 4 cells-11-04084-f004:**
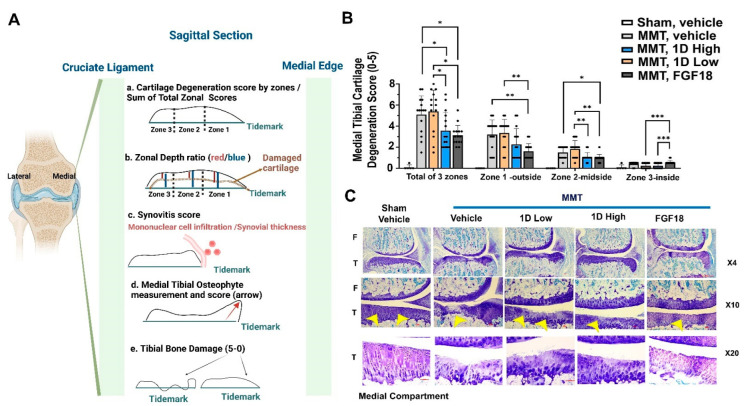
MMT Histopathological Profiling and Cartilage Degenerative scores: (**A**) Exhibits the post-sacrifice histopathological scores employed for the sagittal section of the medial tibial plateau, including (a) cartilage zonal and total degenerative scores; (b) zonal and average depth ratios; (c) synovitis scores; (d) osteophyte measurements and scores; and (e) calcified cartilage (i.e., tibial bone) damage scores. (**B**) Represents cartilage zonal and total degenerative scores for all five experimental groups. These scores exhibit zone-related and cumulative zone-related damage as per the scoring table in Appendix A criteria, and are based on toluidine blue-stained sections (**C**). Representative sections exhibiting dark blue–purple stains depict medial articular cartilage, while light green-blue staining is of connective tissue, bone, and marrow. Medial compartment are shown in magnifications of ×4, ×10, and ×20 for the tibial articular defect site. The left side of the depiction denotes “F” for femur and “T” for tibia. Yellow arrows point towards the cartilage surface, which is intact in Shams, yet damaged in MMT administered with the vehicle. Statistical significance between treatments and control was examined via the Kruskal–Wallis test with a Dunn’s post hoc analysis for scored (non-parametric) parameters, considering *p* < 0.05 (*) to be statistically significant. ** *p* < 0.01; *** *p* < 0.001. Sham, *n* = 5; MMT-Vehicle *n* = 15; MMT-1D, low dose 8 μg *n* = 15; MMT-1D, high dose 24 μg *n* = 15; and MMT-FGF18 3 μg *n* = 15.

**Figure 5 cells-11-04084-f005:**
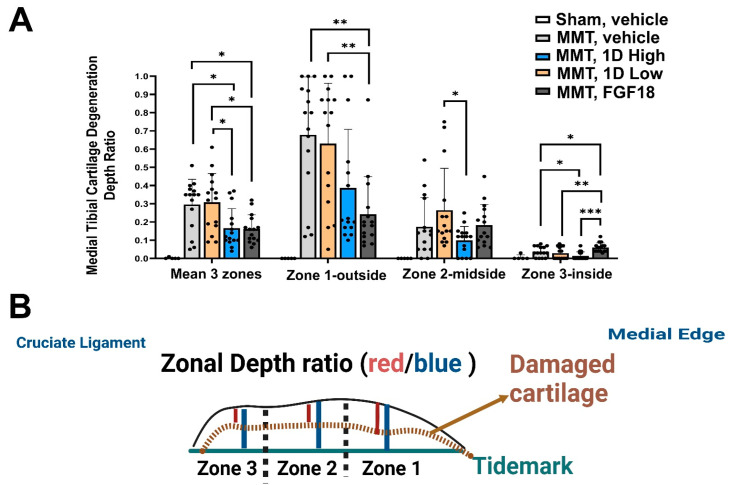
Histopathology for post-MMT Medial Tibial Degeneration Depth ratio. The zonal depth ratio of toluidine-stained sections is shown for all five groups (**A**), per zone of the medial tibial plateau and for the mean of the three zones. (**B**) The lesion depth ratio is calculated by dividing the depth of a zonal lesion (red line from dashed brown line denoting damaged cartilage) by the thickness of articular cartilage from the tidemark (blue line). The deeper the lesion is, the higher the death ratio. Statistical significance between treatments and control was examined via Kruskal–Wallis test with a Dunn’s post hoc analysis for scored (non-parametric) parameters, considering * *p* < 0.05 to be statistically significant. ** *p* < 0.01; *** *p* < 0.001. Sham, *n* = 5; MMT-Vehicle *n* = 15; MMT-1D, low dose 8 μg *n* = 15; MMT-1D, high dose 24 μg *n* = 15; and MMT-FGF18 3 μg *n* = 15.

**Figure 6 cells-11-04084-f006:**
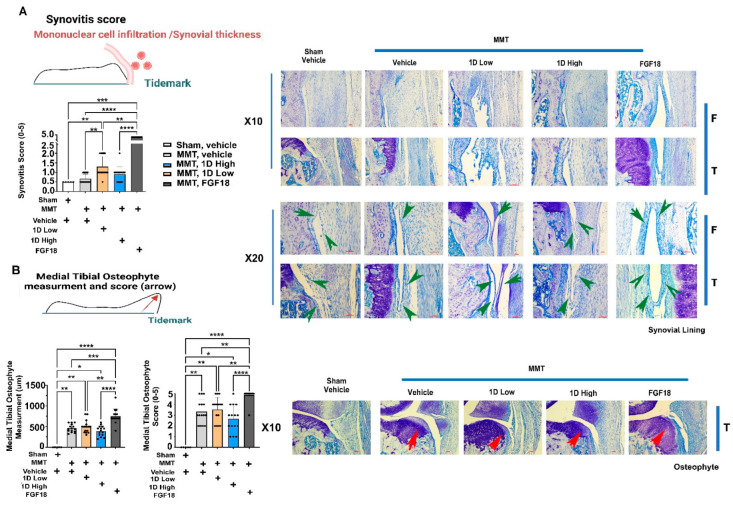
Histopathology for post-MMT Medial Tibial Synovitis and Osteophyte profiles. Synovitis scores were assessed according to the criteria in Appendix A, and displayed for all five groups (**A**) with a scheme and representative images (right panels), including green arrows pointing at the synovial lining. (**B**) Exhibits the measurement of osteophytes (left graph and upper illustration) and the score according to criteria in Appendix A (right graph). The representative images are shown to the left of the graphs in panel B, including red arrows pointing at the measured osteophyte. Statistical significance between treatments and control was examined via the Kruskal–Wallis test with a Dunn’s post hoc analysis for scored (non-parametric) parameters, considering *p* < 0.05 (*) to be statistically significant. ** *p* < 0.01; *** *p* < 0.001; **** *p* < 0.0001. Sham, *n* = 5; MMT-Vehicle *n* = 15; MMT-1D, low dose 8 μg *n* = 15; MMT-1D, high dose 24 μg *n* = 15; and MMT-FGF18 3 μg *n* = 15.

**Figure 7 cells-11-04084-f007:**
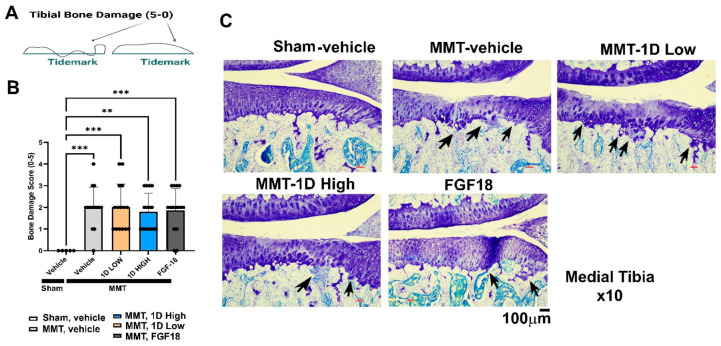
Histopathology for post-MMT Medial Tibial Bone damage. Bone damage is illustrated in (**A**) and exhibited in graphs (**B**). (**C**) exhibits representative sections showing black arrows pointing at the subchondral bone morphology for each group. Statistical significance between treatments and control was examined via Kruskal–Wallis test with a Dunn’s post hoc analysis for scored (non-parametric) parameters, considering to be statistically significant. ** *p* < 0.01; *** *p* < 0.001. Sham, *n* = 5; MMT-Vehicle *n* = 15; MMT-1D, low dose 8 μg *n* = 15; MMT-1D, high dose 24 μg *n* = 15; and MMT-FGF18 3 μg *n* = 15.

## Data Availability

The raw data required to reproduce the above findings are available in the Appendix A. Any additional data may be quired upon requested from corresponding author.

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
