# Peer review of "A Fenchone Derivative Effectively Abrogates Joint Damage Following Post-Traumatic Osteoarthritis in Lewis Rats"

_cells, 2022, doi:10.3390/cells11244084_

Round 1
Reviewer 1 Report
The present manuscript entitled, a Fenchone derivative effectively abrogates joint damage following post-traumatic osteoarthritis in Lewis Rats is a well-designed research study and the authors provided enough experimental evidence to prove their hypothesis and they crosschecked their data.
The abstract part is well-written.
The introduction provided background information and made a solid background for this study.
Materials and methods section clearly describes the experimental procedures that will help researchers reproduce the study easily.
Results section is well-written and well-presented. But still, there is scope to improve. Please make sure all text on the graphs or figures is clearly visible. Figure 4C, please mention each color represents what is in the figures, and indicate the changes (as stated in Figure 6) to make it easy for the readers, if possible, provide enlarged images for better illustration. Please mention each color represents what is in the Figure 6, and enlarge them.
Discussion section is okay.
Please include a separate conclusion based on the data obtained.
Please include a graphical abstract or a summary figure for a better illustration of the study findings.
Please include a limitation section and mention the limitations of the study (what you faced and think that you could do).
Author Response
Reviewer's Comments
Reviewer: 1
Narrative Assessment for Authors:
The present manuscript entitled, a Fenchone derivative effectively abrogates joint damage following post-traumatic osteoarthritis in Lewis Rats is a well-designed research study and the authors provided enough experimental evidence to prove their hypothesis and they crosschecked their data.
General Response: We thank the reviewer for this encouraging feedback and for their efforts in assessing our manuscript. It should be noted that the Rat model was designed according to our published work with a similar CB2 agonist compound (PMID: 36350089) and outsourced to Inotiv, which were not aware of our previous results. It was therefore extremely reassuring to have had similar outcomes in rats, and with another potent CB2 agonist, as these novel Fenchone derivates.
Comment 1: The abstract part is well-written. The introduction provided background information and made a solid background for this study. Materials and methods section clearly describes the experimental procedures that will help researchers reproduce the study easily.
Response 1: We thank the reviewer for this comment. We only made minor revisions for clarity, as required by other reviewers, and in the course of proofreading the text. These are marked in the text and visible via Track changes.
Comment 2: Results section is well-written and well-presented. But still, there is scope to improve. Please make sure all text on the graphs or figures is clearly visible.
Response 2: We thank the reviewer for this comment. It should be noted that these graphs were original graphs from the report supported by Eurofins, regarding specificity and potency of the compounds to CB1 and CB2. We have generated new graphs and included the EC50/IC50 information in the revised Figure 1. Additional information about the curve fit, Max and Min levels are specified in Tables of SD1 and SD2.
Comment 3: Figure 4C, please mention each color represents what is in the figures. …and indicate the changes (as stated in Figure 6) to make it easy for the readers
Response 3: We thank the reviewer for this comment. Figure legend 4C have been amended (Line 367) and citation below (comment 4).
Comment 4: If possible provide enlarged images for better illustration. Please mention each color represents what is in the Figure 6, and enlarge them.
Response 4: We thank the reviewer for this comment. We have included in the figure legend a description of the staining, as mentioned in our above response. Also, we included x20 magnification of medial tibial defect in the articular cartilage, in Fig 4. These additions are updated in the figure legend (Line 398-400);
"(C). Representative sections exhibiting dark blue-purple stain depicting medial articular cartilage, while light green-blue stain-ing is of connective tissue, bone and marrow. Magnifications of the medial compartment are x4, x10 and x20 for the tibial articu-lar defect site. The left side of the depiction denotes "F" for femur and "T" for Tibia. Yellow arrows point towards the cartilage surface, which is intact in Shams, yet damaged in MMT administered with vehicle."
Comment 5: Discussion section is okay. Please include a separate conclusion based on the data obtained.
Response 5: We thank the reviewer for this comment. A conclusion section is added in line 528-538;
“The data displayed in this report support the selective action of Fenchone derivatives on CB2 and their enhanced potency in activating intracellular cAMP levels, as compared to HU308. In particular 1D, exhibited reduced swelling and pain phenotype, partially de-pendent on CB2, using a zymosan inflammatory pain model in mice. Moreover, the structural improvement after IA administration of 1D in a post-traumatic rat model, sup-ports its capacity to prevent joint damage to a comparable degree as FGF18. This structur-al effect of 1D was dose-dependent, but didn’t improve dynamic weight bearing (DWB) after MMT vs vehicle. On the other hand, the effects observed in DWB assays were not worsened upon 1D use vs. FGF18, which served as a DMOAD positive control. Summarizing the cumulative data support that 1D prevented post-traumatic joint dam-age, as well as inflammatory pain, in part by selectively targeting the activation of CB2/cAMP axis.”
Comment 6: Please include a graphical abstract or a summary figure for a better illustration of the study findings.
Response 6: We thank the reviewer for this comment. A graphical abstract is included in the revised version.
Comment 7: Please include a limitation section and mention the limitations of the study (what you faced and think that you could do).
Response 7: We thank the reviewer for this comment. A limitation clause is included in the revised version, line 539-544;
“Limitations of this study include the low number of mice subjected to the zymosan model of inflammatory pain. Moreover, there didn’t appear a significant preference towards 1D for further rat analysis, as both compounds exhibited similar potencies in activating CB2. Another limitation involves pain phenotyping of the MMT model, which could have been extended to other behavioral analysis, as static weigh bearing, Von Frey and thermal plate analysis, to provide a more thorough phenotyping of pain.”
Reviewer 2 Report
I believe that this study makes a significant contribution to the fight against osteoarthritis-related joint inflammation and damage but there are a few minor corrections:
Line 47: Please include the full meaning of CB1/2 in the introduction at first mention.
Line 81-82: Please make use of punctuation marks for better understanding.
Line 164: "24 ug/mL" not "24ug/mL" (make corrections throughout the manuscript = lines 81-83; 102-106; 116; 123; 211; 215; 325-326; 351-352; 377-378; 393-394; 412-413 and figures 2 (A-C) and 3A)
Line 240-248: What statistical test was used for inflammatory pain assessment and serum TNFα?
Line 252: Please remove the question in bracket. I think the right word should be "agonistic" not "agnostic"
Line 253: full meaning of CNR1/2 at first mention.
Line 288-293: The description for figure 2D is not included in the caption
Line 308: full meaning of FGF18 at first mention.
Line 434: Remove duplicate word
Author Response
Reviewer: 2
Narrative Assessment for Authors:
I believe that this study makes a significant contribution to the fight against osteoarthritis-related joint inflammation and damage but there are a few minor corrections:
Response: We thank the reviewer for this encouraging feedback and for their efforts in assessing our manuscript.
Comment 1: Line 47: Please include the full meaning of CB1/2 in the introduction at first mention.
Response 1: Line 66 amended to:
“A previous report shows that endochondral growth plate expresses both cannabinoid re-ceptor 1 (CB1) and cannabinoid receptor 2 (CB2) in the hypertrophic chondrocyte layer and exhibit endocannabinoid tone, which affects limb growth during ,…”
Comment 2: Line 81-82: Please make use of punctuation marks for better understanding.
Response 2: We that the reviewer for this comment. We completely rephrased this section and included appropriate punctuation. Line 103-105:
“As mentioned, vehicle control (1:1:18 ethanol:cremophor:saline) was utilized as placebo. Alternatively, Fibroblast growth factor 18 (FGF18; R&D Systems, catalogue No. 8988-F18-050, lot No. BVE0521071) served as a positive control and administered at 60 mg/mL via IA route.” .
Comment 3: Line 164: "24 ug/mL" not "24ug/mL" (make corrections throughout the manuscript = lines 81-83; 102-106; 116; 123; 211; 215; 325-326; 351-352; 377-378; 393-394; 412-413 and figures 2 (A-C) and 3A)
Response 3: We thank the reviewer for this comment. We have corrected the text and figures accordingly.
Comment 4: Line 240-248: What statistical test was used for inflammatory pain assessment and serum TNFα?
Response 4: We thank the reviewer for this comment. Figure 2 was completely amended after reassessing the statistical analysis and considering the comments from Reviewer 3. The data related to TNF was subsequently removed following our reevaluation of the graph, since the statistics did not qualify for Kruskal Wallis significance. All respective areas in the text were amended accordingly.
Comment 5: Line 252: Please remove the question in bracket. I think the right word should be "agonistic" not "agnostic"
Response 5: We thank the reviewer for noticing this and omitted the text in brackets (line 264). Regarding the agonistic assays (1A) and anatogonistic assays (1B), we included an explanation in the materials and methods (Line 115-131):
“For agonist analysis we monitored “evaluation of potency” (EC50) and %efficacy (Max response), which were reflected by increased intracellular cAMP levels detected in cAMP Hunter™ Gs cell lines (CHO-K1 lines) overexpressing human CNR1 (i.e. gene encoding CB1) or CNR2 (i.e. gene encoding CB2). Alternatively, antagonist activity (i.e. inhibitory concentration or IC50), utilized a cAMP Hunter™ Gi cell lines (CHO-K1 lines) detecting intracellular reduction of cAMP levels in cells overexpressing CNR1 and CNR2. These cell lines are designed to be used in conjunction with the HitHunter® cAMP Assay Detection Kit. For agonist assay, cells were subjected to positive controls, known to activate the increase of cAMP for both receptors (Agonist control 20 µM Forskolin for CB1 and 25µM Forskolin for CB2). For antagonist assay, the cells were initially incubated with an agonist 20 mM or 25mM CP55940, for CB1 or CB2 respectively (SD2). After 30 min of incubation the cells were incubated with vehicle, a range of concentration for antagonist control (i.e. CB1 antagonist AM281 or CB2 antagonist SR144528), or a range of concentrations for 1D/1B fenchones. Prior to testing cell plating media was exchanged with 10uL of Assay buffer (HBSS+10 mM HEPES). Agonist results are expressed as percent efficacy relative to the maximum response of the control ligand (EC50), while antagonist results are expressed as a percent inhibition of the control ligand (IC50) normalized to the unantagonized vehicle.".
Comment 6: Line 253: full meaning of CNR1/2 at first mention.
Response 6: We thank the reviewer for this comment, which requires a clarification. CNR1 and CNR2 are the human genes that encode CB1 and CB2, which are the proteins expressed as cannabinoid receptors on the cell surface. The sentence was amended as below:
Line 117:
”… overexpressing human CNR1 (i.e. gene encoding CB1) or CNR2 (i.e. gene encoding CB2). ".
Line 296:
“compounds to cells either expressing the genes encoding CB1 (i.e. CNR1) or the gene encoding CB2 (i.e. CNR2), as compared ..."
Comment 7: Line 288-293: The description for figure 2D is not included in the caption
Response 7: We thank the reviewer for this comment and request that they refer to our response to comment 4. This data was removed and figure 2 was completely amended after considering the comments from Reviewer 2 and 3.
Comment 8: Line 308: full meaning of FGF18 at first mention.
Response 8: Line 327 amended
Comment 9: Line 434: Remove duplicate word
Response 9: Line 459. “in” removed

Reviewer 3 Report
The authors tested two cannabinoid-fenchone's effects in a rat osteoarthritis model (OA) model. The overall data is not convincing whether 1B/1D are CB2 agonists.
Fig 2: I am not sure from the data that 1B/1D reversed the effects of CB2 antagonist.
Fig 3: Were experiments conducted in mice or rats? Both species are mentioned in the legend.
How was in vivo dose determined?
Why was bone damage measurement not used microCT to image epiphysis?
Fig 6 & 7, arrows should be explained in the legends.
It seems that the manuscript draft was not the final one uploaded, see line 252, where a question has been asked. The table in Fig 1 is confusing; why EC50 given twice?
Author Response
Reviewer: 3
Narrative Assessment for Authors: The authors tested two cannabinoid-fenchone's effects in a rat osteoarthritis model (OA) model. The overall data is not convincing whether 1B/1D are CB2 agonists.
Response: We thank the reviewer for this feedback and for their efforts in assessing our manuscript.
We are also disappointed to hear that the reviewer was not convinced by our research findings. I hope the response below will make some clarifications to better convince the reviewer regarding our conclusion and findings.
The manuscript was built on previous knowledge that was published by Smoum et al., showing the chemical structure and selectivity of several fenchones derivates including 1B/1D (1). Our work attempted to validate these findings, as shown in Figure 1. In particular, 1D and 1B exhibited higher EC50 for cells expressing the CNR2 (encoding CB2) than those expressing CNR1 (encoding CB1). The data in Figure 1A exhibit specify to CB2 for both compounds. The data in the primary human OA-derived chondrocytes (Fig. 1C) also confirmed these data. Notably, our work with another CB2 agonist, HU308, has shown that its effect on chondrocytes tends to upregulate intracellular Camp (2), which is in line with these fenchones derivatives.
On the other hand, Figure 1B, exhibits no antagonist activity, which is defined as reduced cAMP levels due to Gi intracellular activation. This indicates that the response of the cells to Fenchones 1D/1B does not reduce cAMP levels within the cells. We understand that the control in panel 1B with forskolin may have been confusing, as it detects agonistic action wherein cAMP is increased within the cell without subsequent antagonist controls (i.e EC50 in the graphs). We therefore removed the graphs with forskolin in panel 1B, so that this panel may only display the antagonist positive controls displaying IC50 values, which represent reduced cAMP cellular levels. We provide an explanation in the Materials and Methods (line 114).
More importantly, the data in Figure 1A and 1B were carried out independently and blindly by Eurofins, to make sure they reach the same conclusions about these compounds as the report by Smoum et al., (1). We are excited that the outcome of their report was consistent and were also able to confirm it in chondrocytes.
- Smoum R, Haj C, Hirsch S, Nemirovski A, Yekhtin Z, Bogoslavsky B, Bakshi GK, Chourasia M, Gallily R, Tam J, Mechoulam R. Fenchone Derivatives as a Novel Class of CB2 Selective Ligands: Design, Synthesis, X-ray Structure and Therapeutic Potential. Molecules. 2022 Feb 18;27(4):1382.
- Carmon I, Zecharyahu L, Elayyan J, Meka SRK, Reich E, Kandel L, Bilkei-Gorzo A, Zimmer A, Mechoulam R, Kravchenko-Balasha N, Dvir-Ginzberg M. HU308 Mitigates Osteoarthritis by Stimulating Sox9-Related Networks of Carbohydrate Metabolism. J Bone Miner Res. 2022 Nov 9. doi: 10.1002/jbmr.4741. Epub ahead of print. PMID: 36350089.
Comment 1: Fig 2: I am not sure from the data that 1B/1D reversed the effects of CB2 antagonist.
Response 1: We thank the reviewer for this comment. As we wrote in the results section, SR144528 (CB2 antagonist) has on its own an anti-swelling effect, which indeed makes it difficult to determine how well it counteracts the 1D/1B compounds. We reconsidered this comment and decided to omit Figure 2B and justify the use of 1D based on its higher EC50 values and potency, shown in Figure 1A. Accordingly, AM630 (another CB2 antagonist), didn’t display anti-swelling effect at 6h, yet 1D did prevent swelling at 6h. Adding both (1D + AM630) prevented the 1D-related anti-swelling effect completely, at 24h after zymosan stimulation. While the pain effect was not observed at 6h, we could see that adding AM630 to 1D partially reduced the pain sensation after 24h. This indicates that 1D, may act through other pain mechanisms unrelated to CB2.
Line 320:
” As a first step, we utilized a CB2 antagonist, AM630 to block CB2 activity and test if 1D exerts a CB2-dependednt anti-inflammatory and/or anti-pain effect. The data in Figure 2B (Left graph) display reduced swelling at 6h after administering 1D, which was maintained at 24h. Adding AM630 to 1D, was able to reverse the anti-swelling effect of 1D, only 24h after zymosan induction. These data were in line with the von Frey pain phenotypes showing a significant affect for 1D alone, which was partially reversed after blocking CB2 with AM630 at 24h (Fig. 2B, right graph). The pain data imply that 1D may reduce pain sensation, only in part through a CB2 dependent mechanism.”
Comment 2: Fig 3: Were experiments conducted in mice or rats? Both species are mentioned in the legend.
Response 2: We thank the reviewer for this comment. Figure 3 (and thereafter) is with Rats. We have corrected line 360 to include rats.
Comment 3: How was in vivo dose determined?
Response 3: We thank the reviewer for this important remark. We had several considerations when determining the IA doses administered to rats via MMT model;
1) We previously administered HU308 via IA route, at 0.5 mg per joint in mice (PMID: 36350089) with good prophylactic results in a DMM model.
2) The solubility of the fenchone compounds couldn’t exceed 100 mg per joint.
3) Considering an approx. 13-fold higher body weight of adult rats vs mice; and a 20% surplus, brought us to the lowest dose of 8mg per joint; (0.5x13)*1.2=7.8ug. We chose a 3-fold higher dose (24mg), given the solubility restriction and the possible toxicity of higher concentrations, especially given that these concentrations already exceeded the EC50 in chondrocytes.
We added a clarification in the materials
Line 187:
“The fenchone dose was chosen after considering previous injected IA concentration (4) to a mouse joint (0.5 mg) and extrapolating the dose 13-fold to a rats body weight and adding 20% surplus, reaching the lower 8 mg dose administered to the MMT rats”
Comment 4: Why was bone damage measurement not used microCT to image epiphysis?
Response 4: We thank the reviewer for this comment. It is indeed possible to image the subchondral bone via microCT to assess damage, but in this case, we followed the guidelines for histopathology assessment by Gerwin et al., 2010, which is an official OARSI guideline for Rat histopathological analysis. In the future we will consider such additional analysis.
Notably, this particular feature assesses histological presence of basophilia at tidemark, causing loss of calcified cartilage or subchondral bone. Another histological feature of such damage includes the presence of bone marrow lesions or articular cartilage collapse into the epiphysis beneath the tidemark (SD5 scoring table, denoting a range of 0-5, wherein 5 is the most significant level of damage).
Reference:
Gerwin N, Bendele AM, Glasson S, Carlson CS. The OARSI histopathology initiative - recommendations for histological assessments of osteoarthritis in the rat. Osteoarthritis Cartilage. 2010 Oct;18 Suppl 3:S24-34. doi: 10.1016/j.joca.2010.05.030. PMID: 20864021.
Comment 5: Fig 6 & 7, arrows should be explained in the legends.
Response 5: We thank the reviewer for these comments and amended the figure legends 6 and 7 accordingly.
Line 437: “Figure 6. …..(right panels) , including green arrows pointing at the synovial lining. (B) …”
Line 439: “… including red arrows pointing at the measured osteophyte. …"
Line 462: “Figure 7. ….sections showing black arrows pointing at the relevant subchondral bone morphology for each group. ”
Comment 6: It seems that the manuscript draft was not the final one uploaded, see line 252, where a question has been asked.
Response 6: We thank the reviewer for noticing this and omitted the text in brackets, as also noted by Reviewer 2.
Comment 7: The table in Fig 1 is confusing; why EC50 given twice?
Response 7: We thank the reviewer for this comment. We agree the two values were redundant, as one was on a micromolar range and the other on a nanomolar range. We removed the nanomolar range and maintained the micromolar range, for consistency within the other panels within the figure.